# Metacognitive Capabilities of LLMs: An Exploration in Mathematical Problem Solving

**Aniket Didolkar** [1]   **Anirudh Goyal** [1]   **Nan Rosemary Ke** [2]   **Siyuan Guo** [3]   **Michal Valko** [2]   **Timothy Lillicrap** [2]
**Danilo Rezende** [2]   **Yoshua Bengio** [1]   **Michael Mozer** [2]   **Sanjeev Arora** [4]

## Abstract

*Metacognitive knowledge* refers to humans' intuitive knowledge of their own thinking and reasoning processes. Today's best LLMs clearly possess some reasoning processes. The paper gives evidence that they also have metacognitive knowledge, including ability to name skills and procedures to apply given a task. We explore this primarily in context of math reasoning, developing a prompt-guided interaction procedure to get a powerful LLM to assign sensible skill labels to math questions, followed by having it perform semantic clustering to obtain coarser families of skill labels. These coarse skill labels look interpretable to humans.

To validate that these skill labels are meaningful and relevant to the LLM's reasoning processes we perform the following experiments. (a) We ask GPT-4 to assign skill labels to training questions in math datasets GSM8K and MATH. (b) When using an LLM to solve the test questions, we present it with the full list of skill labels and ask it to identify the skill needed. Then it is presented with randomly selected exemplar solved questions associated with that skill label. This improves accuracy on GSM8k and MATH for several strong LLMs, including code-assisted models. The methodology presented is domain-agnostic, even though this article applies it to math problems.

---
*Equal contribution   [1]Mila, University of Montreal   [2]Google Deepmind   [3]The University of Cambridge [4]Princeton University. Correspondence to: Aniket Didolkar <adidolkar123@gmail.com>, Anirudh Goyal <anirudhgoyal9119@gmail.com>.

*The first AI for MATH Workshop at the 41st International Conference on Machine Learning, Vienna, Austria.* Copyright 2024 by the author(s).

## 1. Introduction

Large language models (LLMs) have demonstrated remarkable advancements in recent years at natural language inference tasks ([1–7]), as well as scientific and mathematical problems ([8–11]), although their limitations on mathematical problems are also well-documented ([12–17]).

A core concept in human pedagogy is *Metacognition* ([18]), sometimes described as *thinking about thinking*. It refers to ability to reason about one's own cognitive processes as well as about learning-relevant properties of information or data. *Metacognitive Knowledge* refers to the learner's accumulated knowledge of this type. Pedagogy research shows that improving learners' metacognitive knowledge can improve their capabilities, for example on math ([19; 20]). The current paper raises the question *"Do LLMs also have metacognitive knowledge?"* And if yes, *Can we bootstrap such knowledge to further improve LLM capabilities?*

At first glance, this quest seems difficult. Deciphering LLMs' inner working from their huge set of parameters –all results of non-linear optimization— is notoriously hard. Furthermore, scientists lack parameter access to most leading AI models. But there are still reasons to hope we can understand metacognition by interacting with LLMs. They display some human tics, such as ability to improve their math reasoning via *Chain of Thought (CoT)* ([21]) and also the "Let's think step by step" prompt ([22]). These were generally perceived as convenient tricks to get around the limitations imposed by the LLM's auto-regressive nature. But other pieces of evidence have emerged about existence of LLM metacognition. A notable example is Ask-LLM ([23]), whereby the LLM appears to give surprisingly helpful answers to the question *"Is this a good training datapoint for an LLM?"* The current paper reports on similar direct approach to deciphering LLM metacognition: *Just go ahead and ask it*!

Specifically, the Metacognitive Knowledge of interest in this paper is the catalog of skills (from the LLM's viewpoint) that it applies while solving math questions. Pedagogy research has uncoverex a rich catalog of skills in humans, ranging from simple ones — operations on variables, solving

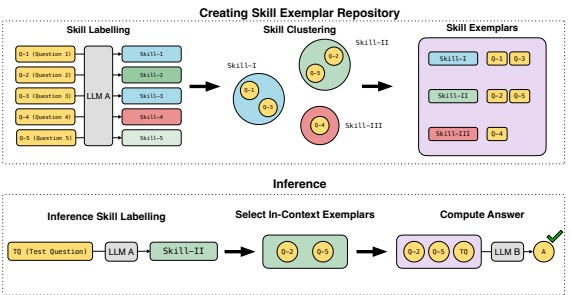

*Figure 1.* **Creating Skill Exemplar Repository**: First, an LLM labels each question with a corresponding skill, as detailed in the prompt provided in Appendix Figure 2 (left). Next, the LLM is asked to combine similar fine-grained skills into broader skill clusters, which represent complex skills. This greatly reduces the number of unique skills from the first stage. The prompt for this is depicted in Appendix Figure 2 (middle). Then the LLM is asked to reclassify all examples from the training set into one of the post-clustering skills. Using these we create a 'Skill Exemplar Repository' which consists of skill exemplars consisting of skill names and their corresponding question/answer examples. **Inference**: During inference, we use an LLM to first label a test question with one of the skills from the skill exemplar repository. Next, we fetch exemplars from the repository with the same skill and provide them as in-context examples to solve the test question.

equations, grasping the concept of a function— to difficult ones such as grasp of difficult theorems and proof strategies. But currently mathematical datasets used in LLM research (such as MATH (16)) partition problems using broad human-assigned topics such as "probability" and "algebra." We are interested in a more fine-grained understanding of LLM skills.

**Skill Discovery:** Our automated approach for the discovery of skills utilizes state-of-the-art LLMs to identify their own catalog of math skills and then organize datasets using that catalog. Stage 1 of our methodology involves instructing the LLM to assign skill labels to each example within a given dataset. Usually this results in fine-grained skills, and too many skill labels. In Stage 2, the same LLM is asked to perform semantic clustering on the labeled data, grouping examples by the similarity of their underlying skills (as perceived by the LLM). Each resulting cluster represents a more coarse-grained skill that is applicable to a larger set of examples. Our method retains only these coarse skills. (To give an example, for the MATH dataset, Stage 1 identified approximately 5000 skills, which Stage 2 reduced to 117 coarse skills.) A random subset of examples representing the coarse skills are retained as its *skill exemplars*. (See Figure 1 and Appendix 8). To subsequently improve (in-context) math problem solving by LLMs we use the repository of skill exemplars –each labeled with a coarse skill. Here the LLM is given a new question and the above list of coarse

skills and asked to identify the skill needed to solve this new question. Then the LLM is provided the previously identified exemplars for the selected skill as in-context examples to guide its problem-solving. We note that this is reminiscent of how human problem-solving is taught by presenting examples very congruent with the specific problem at hand. Here we find that LLM problem solving improves using the skill labels and skill-exemplars provided by an LLM on the same dataset. This provision of skill-exemplars can be seen as a new addition to on top of known prompting methods such as Chain-of-thought.

Although we describe our method only in context of math, it seems general enough to be broadly applicable to problem-solving of other sorts. This is left for future work.

**Paper organization and main results:** Section 3 describes the method and Section 4 describes experiments. Using a strong LLM - GPT-4 - to identify skills, we validate the usefulness of these skills by demonstrates a significant 11.6% enhancement over CoT on the MATH Dataset using the method described in Section 3. Furthermore, the identified skills also improve the generation of code-based solutions for the problems within the MATH dataset giving a 7.52% improvement over the baseline PAL approach (24), which also instructs the model to generate code. Section 4.3 shows that the the skill exemplar repository created for MATH noticeably improved in-context performance for *weaker LLMs* on the same dataset and that the repository for GSM8K helped improve in-context performance for other math datasets. This shows that a powerful LLM can be used for deeper understanding of skills that translates across other LLMs and related datasets.

## 2. Related Works

For human learning, statistical methods can infer latent skills from data and use the inferred skills to more accurately forecast student learning (25; 26). In machine learning, works that study learning via skill induction include (27–30). These start with some definition of skills in terms of model parameters, whereas we use a powerful LLM in a black box way to identify and consolidate skills. A discussion of various prompting strategies is covered in Section 4 and Appendix Section 8.

## 3. Automated Skill Discovery

We describe an automated process for categorizing mathematical questions according to specific *skills* needed to solve them. See Figure 1. Recent works relating skills and LLMs (31; 32) were an inspiration. Conceptually, the strategy involves the creation of a detailed ***skill exemplar*** repository, which contains a compilation of skill names alongside respective illustrative examples (comprising both questions

| Dataset | Topic | Skills |
|---------|-------|--------|
| GSM8K | - | multiplication_and_addition, basic_arithmetic, addition_and_multiplication, arithmetic_operations, multiplication, percentage_calculations, subtraction, algebra, subtraction_and_division, multiplication_and_division, multiplication_and_subtraction, addition_and_subtraction, percentage_calculation, addition_subtraction, average_calculation, subtraction_multiplication, division, addition, linear_equations, algebraic_reasoning |

*Table 1.* **List of Skills for each Dataset** This table lists down the skill obtained after the skill clustering phase for each dataset and corresponding topics. Skill names were provided by GPT-4-0613. The skills of the other topics in MATH can be found in Appendix Table 8

and answers). During the inference stage, when presented with a question, the LLM initially looks among skill exemplars to identify the skill that is best suited for the question. The LLM then utilizes the corresponding exemplars for that skill as in-context prompts.

**Notation.** The proposed setup consist of a training set $\mathcal{T} = \{(q_0^T, a_0^T), (q_1^T, a_1^T), \ldots, (q_n^T, a_n^T)\}$, where $q_i^T$ and $a_i^T$ are question and answers from the training set. The training set is used for selecting in-context examples for inference. Our test set also consists of set of questions and answers - $\mathcal{E} = \{(q_0^E, a_0^E), (q_1^E, a_1^E), \ldots, (q_n^E, a_n^E)\}$. To create the skill exemplars, we first label the training set, $\mathcal{T}$, with a skill per example using a LLM. Next, we label the test set with skills to retrieve in-context examples with matching skills from the skill exemplar repository. The exact procedure of labelling the training and test set with skills is different and we detail both approaches below.

### 3.1. Skill Labelling: categorizing mathematical questions according to specific skills

The process is illustrated in Figure 1. It had the following steps.

**Assign Skill Name for every example in training Set $\mathcal{T}$:** Using a carefully curated prompt (given in Appendix Figure 2 (left)), we asked a LLM to label each training instance with a single skill name and a reason for that assigned skill. Figure 1 (top) represents this process. Applying a strong LLM for this task - GPT-4-0613 - we found that for the $7,000$ instances in the GSM8K dataset (33), it specified approximately 500 unique skill names. For the $7,500$ examples in the MATH dataset (16), it specified $5,000$ skill names. (This perhaps reflects the hardness and diversity of MATH compared to GSM8K.) Although these skill labels precisely encapsulate the capabilities requisite for solving each question, it is clear that the granularity is excessive, raising issues reminiscent of classical "overfitting."

For example, for the question `"In a triangle, the area is numerically equal to the perimeter. What is the radius of the inscribed circle? (A)` 2 `(B)` 3 `(C)` 4 `(D)` 5 `(E)` 6`"` GPT-4 came up with the skill name understanding_of_triangle_properties_and_circle_radius_calculation. Despite descriptive accuracy, its high specificity may limit its utility, as it is improbable that an identical question embodying this precise skill will recur. To address this,

the initial labelling phase is followed by a phase of skill clustering, aiming to generalize the skill categories for broader applicability.

**Semantic Skill Clustering:** In this phase, the LLM was prompted to aggregate the skills identified in the skill labelling stage, specifically to group similar skills into broader categories (Figure 1 (top)) and assign a descriptive label to each category. (The prompt appears in Appendix Figure 2 (middle). ) Again utilizing GPT-4-0613 for this, we obtained a reduced skill set comprising of 22 skills for GSM-8K and 117 skills for MATH. The list of skills are presented in Table 1, and Appendix Table 8. Subsequently, we use the LLM to reclassify all examples in the training set $\mathcal{T}$ using these new skill names from the clustering phase. Thus the initial highly detailed skill labels get consolidated into broader, more universally applicable categories. For instance, the question initially labelled as "understanding_of_triangle_properties_and_circle_radius_calculation" is relabeled to have the skill name "understanding_of_triangles". This modification significantly enhances the applicability of the training set for a wider range of problem-solving scenarios.

**Skill Exemplar Repository:** Following the skill clustering and relabelling process of the training set, we established a 'Skill Exemplar Repository.' This contains a curated selection of skills and their corresponding exemplars, specifically questions and answers, derived from the training set $\mathcal{T}$. The structure of the skill exemplar repository is formalized as follows: skill exemplar repository = $(s_0, q_0^T, a_0^T), (s_1, q_1^T, a_1^T), \ldots, (s_n, q_n^T, a_n^T)$, where $s_i$ denotes the skill label associated with the $i$-th question-answer pair $(q_i^T, a_i^T)$. See Figure 1 (top) for an example of such a repository. This systematic compilation facilitates efficient referencing and application of relevant examples corresponding to specific skills during inference. App. Tables 9, 14, and 15 illustrate examples from the skill exemplar repositories for the GSM-8k and MATH datasets respectively created using GPT-4-0613 . We can see that each question is labelled with a human interpretable and intuitive skill name.

### 3.2. Inference at test time

In the testing phase, the LLM is given a math question $Q$. It is asked to first select one skill from the list of skills in the repository, say $s_i$ that is most relevant to the question.

(The prompt employed for this step appears in Appendix Figure 2 (right).) Next, $K$ Exemplars corresponding to $s_i$, randomly picked, are then employed for few-shot prompting as usual. By providing the LLM with contextually relevant, skill-specific examples from the repository, one expects to enhance its effectiveness at answering the question $Q$. This process is depicted in Figure 1 (below).

**Transferring skill exemplars to other datasets**  The broad range of questions, answers and skill labels in the exemplar repository makes it an attractive source of relevant in-context examples for solving various mathematical problems. To demonstrate such adaptability and utility we applied the Skill Exemplar Repository derived from GSM8K dataset to solving various existing math word problem datasets that were designed to evaluate concrete mathematical skills or concepts. Section 4.3 reports notable improvements in problem-solving capabilities across domains.

### 3.3. Skills from strong LLMs improve weaker LLMs

Through the methodology described above, we find that a strong LLM - such as GPT-4-0613 - is able assign intuitive and human interpretable skill names to questions. These skills are a representation of the metacognitive knowledge of the LLM. We consider whether this knowledge can be applied to other LLMs - specifically weaker LLMs. Section 4.3 shows that skill-based in-context examples, labeled using a stonger LLM as described earlier, also significantly enhance the performance of less advanced models, such as Mixtral (34). This underscores that the skill-based knowledge categorization from one LLM is broadly applicable to other LLMs too.

**Skill-exemplars improve various prompting methods**  Our approach is designed to be synergistic with a range of prompting techniques, thereby offering broad applicability across various methodologies. It can be seamlessly integrated with numerous existing prompting strategies, including the Chain of Thought (CoT) approach (21), PAL (24), and the self-consistency method (35). In each of these instances, the proposed method enhances the existing framework by substituting the conventional in-context examples with those meticulously selected from the Skill Exemplar Repository. This integration not only preserves the inherent strengths of the original prompting techniques but also augments them by leveraging the specificity and relevance of the skill-aligned examples. This adaptability underscores the versatility and potential of the proposed approach to improve the efficacy of various language model prompting strategies.

## 4. Experiments

In Section 3, we have described a procedure to extract metacognitive knowledge from LLMs in the form of skill annotations for mathematical questions. In this section, we show that this knowledge of skills can be further used to improve reasoning in LLMs by using them to provide pertinent in-context examples for solving new mathematical problems through the process described in Section 3.2 and depicted in Figure 1 (below). Our evaluation focused on three distinct areas: *Text-based Prompts*: We utilized chain-of-thought prompting, as detailed in Section 4.1. This method involves providing step-by-step reasoning in the prompt to guide the model's thought process, *Program-based Prompts*: Here, we employed program-aided language models (PALs), described in Section 4.2. PALs integrate programming logic within the language model, aiming to enhance its reasoning capabilities, and *Transferability*: We investigate the generalizability of these skills across different LLMs and datasets, as elaborated in Section 4.3. This aspect tests how well the skills transfer to different LLM models and unseen datasets. Our results demonstrate that knowledge of skills significantly improves performance for both text-based and program-based prompting across different datasets. Furthermore, these skills exhibit strong transferability, boosting mathematical reasoning capabilities across other maths datasets and LLM models. Finally, we conduct a detailed analysis to gain a deeper understanding how our approach influence the reasoning abilities of LLMs.

**Prompting Methods**  We investigate two prominent types of in-context prompting methods for enhancing mathematical reasoning in LLMs: *Text-based Prompting*: Utilizes text examples to demonstrate problem-solving steps, with Chain-of-Thought (CoT) (21) being a prime example. *Program-aided Prompting*: Employs programs to showcase reasoning steps, as seen in Program-aided Language Models (PALs) (24). To assess the effectiveness of these methods, we replaced the standard in-context examples used by CoT (21) and PAL (24) with examples from our skill exemplar repository. We then evaluated the performance of LLMs with both text-based and program-based prompting, using our skill exemplars versus standard examples.

**Baselines**  Our evaluation also includes a comparison with four baselines to isolate the impact of our skill-specific examples: *Random*: This baseline randomly selects examples from our repository in contrast to CoT's fixed examples, highlighting the necessity of skill-aligned example selection. *Topic-Based*: Examples are grouped by broader mathematical topics (e.g., algebra), as in the MATH dataset (16). This tests whether finer-grained skills (as detailed in Table 8) offer an advantage over broader topic categorizations. *ComplexCOT* (36): Chooses complex in-context examples for CoT, allowing us to analyze whether complexity or skill-

| Prompting | Pre-Algebra | Geometry | Inter-Algebra | Algebra | Probability | Pre-Calculus | Num. Theory | Overall |
|---|---|---|---|---|---|---|---|---|
| CoT | - | - | - | - | - | - | - | 42.2 |
| Complex CoT | 71.6 | 36.5 | 23.4 | 70.8 | 53.1 | 26.7 | 49.6 | 50.30 |
| CoT + Topic-Based | 71.16 | 39.45 | 24.14 | 67.90 | 54.64 | 31.13 | 47.03 | 50.31 |
| CoT + Skill-Based | 74.28 | 41.75 | 27.02 | 73.12 | 58.01 | 33.70 | 51.10 | 53.88 |

*Table 2.* **Text-based prompt results on the MATH Dataset.** Our Skill-Based approach, employing CoT prompting, demonstrates superior performance over all other methods across all topics within the MATH dataset. All experiments were conducted using GPT-4-0613.

specificity has a greater impact on performance. *Retrieval-RSD* (37): This selects relevant in-context examples for few-shot tasks similar to the proposed approach. They first map the examples to a latent space and then selects top-k in-context examples based on cosine similarity to the example. Through these comparisons, we aim to discern the relative benefits of skill-specificity and complexity in example selection for enhancing LLMs' mathematical reasoning capabilities.

**Datasets** We evaluate the proposed approach using a variety of mathematical reasoning datasets. We start with the GSM8K dataset (33), which comprises grade-school level math problems. We then move on to the challenging MATH dataset (16), known for its competition-level problems.

To examine the transferability of skills, we apply the skills from the GSM8K dataset to other math word problem datasets. These include SVAMP (15), ASDIV (38), and the MAWPS suite (SingleOP, SingleEQ, AddSub, MultiArith) (39). Each dataset presents its unique set of challenges and complexities, allowing us to thoroughly assess the adaptability and effectiveness of our approach across different mathematical contexts. For details about these datasets, please refer to the Appendix 9.1.

**Language Models** In Section 9.4 of the Appendix, we conduct a comparative analysis of GPT-4-0613, GPT-3.5-Turbo, and Mixtral-8x7B in their proficiency in generating precise skill labels. Through experimentation, we show that the skill labels annotated by GPT-4-0613 lead to the strongest in-context learning performance on the MATH dataset (16). Therefore, we establish GPT-4-0613 as the primary model for skill labeling, clustering, and conducting the majority of our experiments. For transfer experiments, as outlined in Section 3.3 and further detailed in Section 4.3, we evaluate the performance of the Mixtral 8x7B model (34). This dual-model approach allows us to assess the effectiveness of our methods across different advanced language models.

### 4.1. Text-based Prompts

We consider the GSM8K dataset (33), containing grade-level math word problems, and the MATH dataset (16), featuring competition-level math problems. These experiments aim to assess the efficacy of our approach across a wide range of mathematical complexities, specifically using text-based prompting strategies. All experiments were

| Base Model | Prompting | GSM8K |
|---|---|---|
| GPT-3.5-Turbo | Retrieval RSD | 76.8 |
| | CoT + Skill-Based | 82.03 |
| GPT-4-0613 | CoT | 93.00 |
| | CoT + Random | 92.87 |
| | CoT + Skill-Based | 94.31 |
| | CoT + Skill-Based (maj@5) | 95.38 |

*Table 3.* **Text-based prompt results on the GSM8K Dataset.** Our Skill-Based approach outperforms various other methods on the GSM8K dataset across two different models: GPT-3.5 Turbo and GPT-4-0613. Refer to text for description of baselines.

carried out using GPT-4-0613, employing 8-shot prompting and a decoding temperature set to 1.0.

**Results on GSM8K.** GSM8K dataset (33) contains 7.5k training problems and 1k test problems. The skill exemplar repository is created using the training data only, refer to Section 3.1 for details. See Table 9 in the appendix for examples from the skill exemplar repository.

We utilize the skill exemplar repository to solve test set problems from the GSM8K dataset, as outlined in Section 3.2. The results are shown in Table 3. Our Skill-Based approach outperforms both the Chain-of-Thought (CoT) and Random baselines on the GSM8K dataset, underscoring the importance of accurate skill assignment and pertinent in-context examples in effective problem-solving. Furthermore, augmenting the Skill-Based approach with self-consistency (SC, presented as maj@5 in Table 3) techniques (35) leads to even better performance, highlighting the adaptability and effectiveness of our method. For the SC experiments, we sample 5 reasoning chains from the LLM and choose the most frequent answer. Additionally, we provide a detailed breakdown of per-skill accuracy for both the proposed approach and the Random approach in Appendix Figure 3. To further emphasize the effectiveness of the proposed method, we compare it to the Retrieval-RSD approach (37) which is also a pertinent in-context example selection approach for few-shot prompting. The results are presented in Table 3 show the superiority of our approach as compared to the Retrieval-RSD approach. We use GPT-3.5-Turbo backbone for this comparison.

**Results on MATH.** The MATH dataset, comprising competition-level math problems, covers topics like Pre-Algebra, Algebra, Intermediate Algebra, Geometry, Number Theory, Precalculus, and Probability. Its training set has 7.5k examples and the test set has 5k examples, each labeled by their respective topics. Following the methodology

described in Section 3.1, we created a Skill Exemplar Repository using the MATH dataset's training set. This repository is showcased through examples in Appendix Tables 14 and 15, providing insights into the range and nature of skills covered in the MATH dataset. Furthermore, in Appendix Table 10 we show examples of the relevant in-context examples selected from the skill exemplar repository to solve a given question. We can see that selected exemplars are similar to the question to be and correctly illustrate the concepts required by the question.

Results on the MATH dataset are shown in Table 2. For this analysis, our proposed approach utilizes a straightforward Chain-of-Thought (CoT) method, wherein the in-context examples are sourced from the skill exemplar repository. Our method achieves a notable improvement in performance, surpassing the standard Chain-of-Thought (CoT) by an impressive 11.6%. We also outperform 3.5% over Complex CoT, and 3.5% over the Topic-Based approach. These results highlight the efficacy of our approach, particularly with its fine-grained skill labeling. The fact that it surpasses Complex CoT is especially noteworthy, indicating the importance of selecting in-context examples that are highly relevant to the specific problem, rather than using their complexity as a proxy for quality.

### 4.2. Program-Aided Prompts

Program Aided Language Models (PALs), as developed by 24, are designed to produce program-based solutions for problem-solving, employing Python as the programming language. This approach addresses the issue of logical or calculation errors in Language Models by translating solutions into code, thus leveraging a compiler for final answer generation. In our integration of Skill-Based prompting with PALs, we modify the in-context example structure: we use three non-code-based examples from our skill exemplar repository based on skill matching, followed by one fixed code-based example, totaling four in-context examples. The specifics of these prompts are detailed in Appendix Section 9.5.

Our experiment, results of which are shown in Table 4, tests this modified approach on a subset of 500 examples from the MATH test set, same as those used in (40). The findings are significant: despite only one code-based in-context example (compared to PAL's four), our approach shows a 7.52% improvement over PAL. This highlights the impact of strategically chosen, relevant in-context examples, on the accuracy of code generation for problem-solving.

### 4.3. Transfer of Skill Exemplars

**Studying Transfer of Skill Exemplars to Other Models**
In this study, we explore the transferability of skills from GPT-4 to other LLMs, particularly focusing on Mixtral

8x7B (34). This experiment replicates the setup used for the earlier experiments with GPT-4-0613, utilizing the same skill exemplar repository and skill labels for the MATH dataset test set examples initially labeled by GPT-4-0613. For each problem, 4 in-context examples are chosen based on skill-matching, and outputs are sampled with a decoding temperature of 0.2. The results are displayed in Table 5. We use 1 A100L GPU for this experiment.

Here, we compare our Skill-Based approach against two baselines: Chain-of-Thought with self-consistency (SC) as per (35) and the Topic-Based approach. For implementing self-consistency, we generate four reasoning chains and select the most frequent answer (noted as maj@4 in Table 5). The results demonstrate that our Skill-Based approach surpasses both the Topic-Based and CoT approaches. Notably, our approach, even without self-consistency, matches the performance of CoT with SC, highlighting its efficacy in extracting correct reasoning paths and concepts. Furthermore, when combined with self-consistency, our approach shows a remarkable 4.0% improvement over CoT with SC, affirming its superior efficacy in skill application and reasoning.

**Studying Transfer of Skill exemplars to Other Datasets**
Here, we investigate the transferability of skills from the GSM8K training dataset to other math word problem datasets. We apply our approach to various datasets, including SVAMP (15), ASDIV (38), SingleOP, SingleEQ, AddSub, and MultiArith (39), each comprising distinct problem types. We utilize the GSM8K-derived skill exemplar repository for these datasets, testing skill transferability across similar datasets. Notably, we use the pre-clustering skill labels, as these datasets feature finer granularity problems compared to GSM8K, making post-clustering skills less effective.

The results, presented in Table 6, demonstrate the effectiveness of our approach. We employ a CoT-based method with 4-shot prompting and greedy decoding, aligning with the baseline settings. Our Skill-Based approach consistently surpasses the base CoT across all datasets. We also benchmark against a PAL-based approach and a hybrid CoT + PAL approach from (41), where the model outputs both CoT and PAL solutions and selects the most accurate. Our Skill-Based approach outperforms CoT + PAL in 4 out of 6 datasets, offering a simpler yet more effective solution. These findings affirm the potential of skill knowledge transfer from one dataset to other similar datasets.

### 4.4. Analysis

We delve into the impact of Skill-Based on precise concept and skill application, Firstly, we pinpoint successful instances where Skill-Based prompts guide the LLM in selecting and applying the correct skills. Secondly, we investigate cases where, despite pertinent Skill-Based prompts, the

| Prompting | Inter Algebra | Precalculus | Geometry | Num. Theory | Probability | PreAlgebra | Algebra | Overall |
|---|---|---|---|---|---|---|---|---|
| PAL (4-shot) | 30.9 | 23.2 | 31.7 | 66.1 | 57.9 | 73.2 | 65.3 | 52.0 |
| PAL + Skill-Based (3 Skill-Based + 1 Code-Based) | **35.05** | **44.64** | **39.02** | **70.97** | **60.53** | **78.05** | **72.58** | **59.32** |
| PAL + Skill-Based (7 Skill-Based + 1 Code-Based) | 37.11 | 53.57 | 41.46 | 72.58 | 65.79 | 81.70 | 73.39 | 62.00 |

*Table 4.* **Program-aided prompts results on the MATH dataset.** This table illustrates the performance achieved by employing the Skill-Based approach to generate code for problem-solving tasks drawn from the MATH dataset using **GPT-4-0613**. Evidently, supplying pertinent in-context examples grounded in specific skills enhances the program generation performance of GPT-4-0613, leading to a notable improvement across all topics encompassed in the MATH dataset.

| Prompting | SC (maj@n) | Pre Algebra | Geometry | Inter- Algebra | Algebra | Probab- ility | Pre- Calculus | Num. Theory | Overall |
|---|---|---|---|---|---|---|---|---|---|
| CoT | maj@4 | - | - | - | - | - | - | - | 28.4 |
| + Topic-Based | × | 42.94 | 17.33 | 11.30 | 40.78 | 19.83 | 14.47 | 16.85 | 26.14 |
| + Skill-Based | × | **47.76** | **19.42** | **13.29** | **43.05** | **20.04** | **16.12** | **18.33** | **28.44** |
| + Topic-Based | maj@4 | 52.58 | 20.25 | 10.68 | 48.78 | 24.05 | 14.65 | 20.93 | 30.75 |
| + Skill-Based | maj@4 | 53.96 | 22.55 | 13.68 | 49.70 | 24.26 | 18.32 | 21.48 | 32.44 |

*Table 5.* **Transfer Skill Exemplars to Other Models.** All experiments are performed using the MATH dataset on the **Mixtral 8 × 7B** model, comparing against standard CoT, CoT with topic-based exemplars, CoT with skill-based exemplars, CoT with self-consistency (maj@4) using both topic and skill-based exemplars. Skill labels and exemplars are obtained from GPT-4-0613. The enhanced performance of Skill-Based indicates effective transferability of skills from GPT-4 to another model.

| Prompting | SVAMP | SingleOP | SingleEQ | AddSub | MultiArith | ASDIV |
|---|---|---|---|---|---|---|
| CoT | 91.9 | 97.2 | 97.2 | 93.9 | 98.0 | 92.7 |
| PAL | 92.2 | 95.2 | 96.8 | 94.9 | 98.5 | 90.2 |
| CoT + PAL | 93.7 | 97.3 | 98.6 | 95.7 | 99.0 | 93.5 |
| CoT + Skill-Based | 92.6 | 97.86 | 99.01 | 96.71 | 98.17 | 94.03 |

*Table 6.* **Transfer of Skill Exemplars to Other Datasets** Investigation of skill transfer from GSM8K to different math word problem datasets using GPT-4-0613. Questions in the target dataset are labeled with corresponding skills from GSM8K, and in-context examples are selected based on skill-matching. The proposed approach achieves the highest accuracy in 4 out of 6 cases.

**LLM fails to utilize the right concepts.** Lastly, we quantify these instances of failure and compare them against baseline models, assessing the efficacy of Skill-Based prompting in enhancing the LLM's performance. All experiments are performed using GPT-4-0613.

**Instances of LLM benefiting from Skill-Based Approach** In Table 7, we compare the effectiveness of the Skill-Based approach against the Topic-Based approach in problem-solving scenarios through examples. The Skill-Based approach significantly improves the model's reasoning and skill application. We highlight the reasoning errors of the Topic-Based approach in red and the correct reasoning steps undertaken by the Skill-Based approach in blue.

Our analysis reveals that the Topic-Based approach misapplies essential skills. For example, Table 7 shows a fundamental misunderstanding of trigonometry in Example 1 and fails to recognize negative cubes in Example 2. These errors are notably absent in the Skill-Based approach, demonstrating its superior understanding and application of key concepts.

**Occurrences of Incorrect Answers Despite Employing a Skill-Based Approach** We examine the limitations of the skill-based approach in Table 13 (appendix). This table

highlights instances where the model, despite using a skill-based approach, fails to produce correct answers. We use blue to denote correct reasoning steps and red for errors.

In Example 1, both the Skill-Based and Topic-Based approaches correctly apply the logarithm formula but err in selecting the appropriate number or input, categorizing this error as a "main skill error" or "skill error." This demonstrates a failure in correctly applying the primary skill needed for the question, highlighting a limitation of the proposed approach. Example 2 further illustrates this limitation. Although the Skill-Based approach correctly uses counting concepts, it erroneously calculates the number of diagonals in a hexagon. This error indicates a shortfall in the application of certain secondary skills required to solve the problem such as, in this instance, understanding properties of a hexagon.

These examples suggest that while the Skill-Based approach effectively guides the application of the main skill required for a question, it may falter in the application of secondary skills or in the comprehension of specific question properties. This analysis underlines the approach's strengths in primary skill application but also its limitations in more nuanced or compound skill scenarios. It would be worthwhile to work with more complex skills.

**Additional Metrics** We introduce three metrics to evaluate the effectiveness of the proposed approach, using examples from the MATH dataset and employing GPT-4-0613 for classification. These metrics are: MAIN SKILL ERROR (SKILL ERROR): This indicates a failure in understanding or applying the primary skill required for a question, SECONDARY SKILL ERROR: This denotes errors in comprehending or applying secondary skills necessary for the question, CALCULATION ERROR: This reflects mistakes in the calculation

|  | Example 1 | Example 2 |
|---|---|---|
| **Question** | *Compute* $\sin 210^\circ$. | *Find* $\left\lceil \sqrt[3]{-25} \right\rceil$ |
| **Topic-Based Solution** | To solve the problem, we have to know that the trigonometric functions have period $360^\circ$, which means $\sin 210^\circ = \sin(210^\circ - 180^\circ) = \sin 30^\circ = 1/2$. | The cube root of $-25$ is not a real number, hence it cannot be evaluated. The ceiling function is applicable only to real numbers, therefore this expression is undefined. |
| **Skill-Based Solution** | The angle $210^\circ$ is $30^\circ$ below the negative $x$-axis, so $\sin 210^\circ = \sin(-30^\circ) = -1/2$. **Skill Used:** trigonometry. | The cube root of -25, $\sqrt[3]{-25}$, equals -2.92 approximately, since the ceiling function rounds up to the nearest integer greater than the given number, the value will be $-2$. **Skill Used:** exponent_and_root_skills |

*Table 7.* **Instances Benefiting from Skill-Based Approach** This table illustrates instances where our skill-based approach empowers the Language Model (LLM) to apply relevant skills effectively. Red-highlighted text reveals conceptual errors by the Topic-Based baseline, while blue-highlighted text showcases skillful and accurate skill application.

process during question-solving.

These error types are not mutually exclusive; a single instance may exhibit multiple error types. Correctly solved instances show none of these errors. GPT-4-0613's role in classifying examples into these categories is detailed in Appendix, Section 9.7, and its effectiveness is evidenced by the classifications in Table 13. To calculate the metrics, we first determine error rates for each error type and then derive success rates. These rates indicate how often the model correctly applies main and secondary skills, as well as performs calculations, across various questions.

Appendix Figure 4 displays the SKILL SUCCESS RATE, SECONDARY SKILL SUCCESS RATE, and CALCULATION SUCCESS RATE for both Skill-Based and Topic-Based approaches. We expect the skill-based in-context example selection to be useful for reducing main skill errors. Our hypothesis is supported by our findings, which show a higher SKILL SUCCESS RATE for this approach. This suggests that the model more frequently uses the correct skill with the Skill-Based approach compared to the Topic-Based baseline. Additionally, the proposed approach also demonstrates effectiveness in reducing secondary skill errors and calculation errors, underscoring its overall superior performance.

## 5. Discussion and Conclusion

We presented a framework for extracting metacognitive knowledge from Large Language Models in the form of skills that categorize questions in mathematical datasets based on concepts required to solve them. This led to a Skill Exemplar Repository, containing a list of mathematical question-answer pairs annotated with the respective skills needed (in the LLM's own estimation) for the solution. Leveraging this repository, we furnish pertinent in-context examples to Large Language Models (LLMs) for tackling previously unseen mathematical questions. Our experiments show substantial empirical enhancements across diverse mathematical datasets, ranging from grade-level math problems to intricate competition-level math challenges. The enhancements in performance via use of the repository also

transfers to weaker LLMs.

One limitation of our methodology is that it assigns only one skill to each math question. As discussed in Section 4.4, mathematical problems often require a combination of a primary skill and various secondary skills. We leave design of a more advanced approach —say, using an LLM to create hierarchies of skills to assign multiple skills to each datapoint— for future work.

While this paper primarily addresses in-context learning, our future goal is to extend these methodologies to improve all models through fine-tuning processes. Presently, our framework relies on the availability of advanced models like GPT-4. However, the skill discovery process improved in-context learning for GPT-4, which suggests that using skills to fine-tune GPT-4 may raise its capabilities. This hints more broadly at a path towards bootstrapping model capabilities –and not just in math—that seems worth exploring.

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

| Dataset | Topic | Skills |
|---|---|---|
| GSM8K | - | multiplication_and_addition, basic_arithmetic, addition_and_multiplication, arithmetic_operations, multiplication, percentage_calculations, subtraction, algebra, subtraction_and_division, multiplication_and_division, multiplication_and_subtraction, addition_and_subtraction, percentage_calculation, addition_subtraction, average_calculation, subtraction_multiplication, division, addition, linear_equations, algebraic_reasoning |
| MATH | Pre-Algebra | average_calculations, ratio_and_proportion, geometry, basic_arithmetic_operations, fractions_and_decimals, probability_and_combinatorics, multiplication_and_division, counting_and_number_theory, prime_number_theory, multiples_and_zero_properties, solving_linear_equation, circles, exponentiation_rules, perimeter_and_area |
| | Algebra | combinatorial_operations_and_basic_arithmetic, function_skills, calculation_and_conversion_skills, solving_equations, inequality_skills, graph_and_geometry_skills, number_theory_skills, factoring_skills, complex_number_skills, sequence_and_series_skills, quadratic_equation_skills, geometric_sequence_skills, polynomial_skills, ratio_and_proportion_skills, logarithmic_and_exponential_skills, algebraic_manipulation_skills, distance_and_midpoint_skills, arithmetic_skills, exponent_and_root_skills, algebraic_expression_skills, function_composition_skills |
| | Inter-Algebra | solving_inequalities, understanding_and_application_of_functions, inequality_solving_and_understanding, quadratic_equations_and_solutions, calculus_optimization_skills, polynomial_skills, understanding_and_applying_floor_and_ceiling_functions, summation_and_analysis_of_series, function_composition_and_transformation, sequence_and_series_analysis_skills, solving_system_of_equations, understanding_and_utilizing_infinite_series, recursive_functions_and_sequences, complex_number_manipulation_and_operations, understanding_ellipse_properties, complex_numbers_related_skills, simplification_and_basic_operations, graph_understanding_and_interpretation, understanding_logarithmic_properties_and_solving_equations, understanding_and_manipulation_of_rational_functions, properties_and_application_of_exponents, algebraic_manipulation_and_equations, prime_number_recognition_and_properties, absolute_value_skills |
| | Geometry | understanding_circle_properties_and_algebraic_manipulation, other_geometric_skills, pythagorean_skills, quadrilateral_and_polygon_skills, triangle_geometry_skills, calculus_skills, 3d_geometry_and_volume_calculation_skills, circle_geometry_skills, area_calculation_skills, coordinate_geometry_and_transformation_skills, ratio_and_proportion_skills, trigonometry_skills, combinatorics_and_probability_skills, algebraic_skills |
| | Number Theory | base_conversion, prime_number_theory, greatest_common_divisor_calculations, modular_arithmetic, solving_equations, number_theory, factorization, division_and_remainders, exponentiation, sequence_analysis, arithmetic_sequences, basic_arithmetic, polynomial_operations, understanding_of_fractions, number_manipulation |
| | Precalculus | matrix_operations, geometric_series_comprehension, basic_trigonometry, vector_operations, coordinate_systems, trigonometric_calculations, complex_numbers, geometric_relations, calculus, algebra_and_equations, three_dimensional_geometry, arithmetic_operations, parametric_equations, sequences_series_and_summation, geometry_triangle_properties, geometry_and_space_calculation, determinant_calculation, geometry_transforms, complex_number_operations |
| | Probability | probability_calculation_with_replacement, combinatorics_knowledge, probability_theory_and_distribution, combinatorial_mathematics, counting_principals, permutation_and_combinations, probability_concepts_and_calculations, calculating_and_understanding_combinations, number_theory_and_arithmetic_operations, factorials_and_prime_factorization, understanding_and_applying_combinatorics_concepts |

*Table 8.* This table lists down the skill obtained after the skill clustering phase for each dataset and corresponding topics.

# Appendix

## 6. List of Skills

In this section, we list down the skills that make up the skill exemplar repository for each of the GSM8K and MATH Datasets after the skill clustering phase.

## 7. Prompts Used for Skill Labelling and Skill Clustering

This section presents the prompts used for labelling the skills from the training set $\mathcal{T}$ and the test set $\mathcal{E}$ as well as the prompt used for clustering the skills. The training set skill labelling prompt is shown in Figure 2(left), the skill clustering prompts is shown in Figure 2 (middle), and the test set skill labelling prompt is shown in Figure 2 (right).

## 8. Related Works: Prompting Strategies

**Prompting Methods** Prompting methods help enhance the reasoning abilities of language models. *Chain-of-Thought (CoT) prompting*, 21, provides in-context math questions together with solutions which include detailed reasoning chains. *Program-Aided Language Models (PAL)*, 24, instruct the model to produce a code-based solution to the given problem by providing in-context examples that also contain code-based solutions. *Ensemble methods*, based on CoT and PAL (35; 42), incorporate self-consistency, where the most frequent answer is chosen (35), and *progressive-hint-prompting*, which utilizes

**Input Prompt:**
Consider this mathematical question. Label this question with a mathematical skill that would be required to solve the question. Basically, you should be able to use the skill as a dictionary key in python. The skill name should be lower case letters only. The skill name should be very descriptive and you may use multiple words to describe the skills required in the question. If you do use multiple words per question, then join them by an underscore .

question: A printing press is printing brochures. The press prints 20 single-page spreads, and twice as many double-page spreads. For each 4 pages printed for the spreads, the press prints a block of 4 ads, each of which take up a quarter of a page. The brochures can be arranged in any order as long as they are made up of 5 pages each. How many brochures is the printing press creating?

Your answer should be as follows:
<name of the skill>, reason: <reason for the skill>

**GPT 4 Response:**
multi_step_problem_solving
reason: The problem involves many steps - first finding the total number of pages printed, then how many of those pages are ads or spreads, and finally determining how many 5-page brochures can be made from the total.

**Input Prompt:**
Here is a list of skills required to solve a mathematical question:

[average_calculation, proportion_calculation, percentages, order_of_operations, subtraction, addition, multiplication, addition_of_fractions, permutations_combinations, counting, understanding_fractions_and_inequalities]

Reduce the number of unique skills by grouping similar skills into categories and give a descriptive name to each category.

**GPT 4 Response:**
1. basic_arithmetic_operations: addition, subtraction, multiplication

2.fractions_and_percentages: addition_of_fractions, understanding_fraction_and_inqualities, percentages

3. average_and_proportion: average_calculation, proportion_calculation

4. order_of_operations: order_of_operations

5: combinatorics_and_counting: counting, permutations_combinations

**Input Prompt:**
Here is a list of skills required to solve a mathematical question:

[s_0, s_1, s_2, ...., s_n]

Here are some examples of questions that demonstrate how these skills are applicable:

question: <question_0> skill: s_0
question: <question_1> skill: s_1
question: <question_2> skill: s_2
.
.
.
question: <question_n> skill: s_n

Label the new question with one skill from the list.

question: <question>

Your answer should be as follows:
<name of the skill>, reason: <reason for the skill>

**GPT 4 Response:**
s_i, reason: <reason>

*Figure 2.* **Prompts for Skill Labelling and Clustering** (left) The prompt which is used for labelling all examples in the training set $\mathcal{T}$ with skills. (middle) The prompt used for clustering the skills obtained after skill labelling. (right) The prompt used to label each test set example with skills.

a feedback-based strategy for refining responses (42). Notably, all these methodologies employ a fixed set of in-context examples. A strategy for selecting in-context examples was introduced in ComplexCoT (36), which prefers in-context examples of higher complexity, i.e., length of the reasoning chains. Our approach proposed also provides dynamically selected in-context examples sourced from the Skill Exemplar Repository. In our case, examples are selected based on relevance rather than complexity. The proposed approach can seamlessly integrate with any of the above prompting methods.

# 9. Experimental Details

## 9.1. Description of Datasets

- **GSM8K Dataset** (33) - This dataset consists of 7.3k math word problems in the training set and 1.3k math word problems in the test set.

- **SVAMP Dataset** (15) - This dataset consists of 1k grade 4 and lower level math word problems but they introduce certain variations in each problem making it more challenging for LLMs to solve.

- **ASDIV Dataset** (38) - This is a dataset consisting 2.3k grade level math word problems. It contains a lot of diversity in terms of language patterns and types of problems considered.

- **Single EQ Dataset** (39) - This dataset consists of 509 single equation word problems.

- **Single OP Dataset** (39) - This dataset consists of 562 single operation math word problems.

- **AddSub Dataset** (39) - This dataset consists of 295 addition and subtraction word problems.

- **MultiArith Dataset** (39) - This dataset consists of 600 multi-step arithmetic problems.

- **MATH Dataset** (16) - This dataset consists of 7.5k training and 5k test competition-level math problems. They cover the following mathematical topics - Pre-Algebra, Algebra, Intermediate Algebra, Pre-Calculus, Geometry, Number Theory, and Probability.

| Skills | Questions | Answers |
|---|---|---|
| proportional_reasoning | Weng earns 12 an hour for babysitting. Yesterday, she just did 50 minutes of babysitting. How much did she earn? | Weng earns 12/60 = 12/60=0.2 per minute. Working 50 minutes, she earned 0.2 x 50 = 0.2*50=10. |
| percentage_calculations | Mark has a garden with flowers. He planted plants of three different colors in it. Ten of them are yellow, and there are 80% more of those in purple. There are only 25% as many green flowers as there are yellow and purple flowers. How many flowers does Mark have in his garden? | There are 80/100 * 10 = 80/100*10=8 more purple flowers than yellow flowers. So in Mark's garden, there are 10 + 8 = 10+8=18 purple flowers. Purple and yellow flowers sum up to 10 + 18 = 10+18=28 flowers. That means in Mark's garden there are 25/100 * 28 = 25/100*28=7 green flowers. So in total Mark has 28 + 7 = 28+7=35 plants in his garden. |
| fraction_calculation | Lisa, Jack, and Tommy earned $60 from washing cars all week. However, half of the $60 was earned by Lisa. Tommy earned half of what Lisa earned. How much more money did Lisa earn than Tommy? | Lisa earned $60 * 1/2 = $60*1/2=30. Tommy earned $30 * 1/2 = $30*1/2=15. Lisa earned $30 - $15 = $30-15=15 more than Tommy. |
| volume_calculation | Nancy is filling an aquarium for her fish. She fills it halfway and goes to answer the door. While she's gone, her cat knocks the aquarium over and spills half the water in it. Then Nancy comes back and triples the amount of water in the aquarium. If the aquarium is 4 feet long, 6 feet wide, and 3 feet high, how many cubic feet of water are in the aquarium? | First calculate the volume of the aquarium by multiplying its length, width and height: 4 ft * 6 ft * 3 ft = 4*6*3=72 cubic ft Then figure out what proportion of the aquarium is full after the cat knocks it over: 1/2 * 1/2 = 1/4 Then figure out what proportion of the aquarium is full after Nancy refills it: 3 * 1/4 = 3/4 Now multiply the proportion of the aquarium that's full by the aquarium's volume to find out how much water is in it: 72 cubic ft * 3/4 = 72*3/4=54 cubic ft |

*Table 9.* This dataset shows examples from the skill exemplar repository constructed using the GSM8K training dataset.

| Question | In-Context Q1 | In-Context Q2 | In-Context Q3 |
|---|---|---|---|
| Compute $\sin 210°$ | Compute $\sin 510°$. | Compute $\tan(-3645°)$. | Find $\tan Y$ in the right triangle shown below.[asy] pair X,Y,Z; X = (0,0); Y = (24,0); Z = (0,7); draw(X–Y–Z–X); draw(rightanglemark(Y,X,Z,23)); label("$X$",X,SW); label("$Y$",Y,SE); label("$Z$",Z,N); label("25",(Y+Z)/2,NE); label("24",Y/2,S); [/asy] |
| Find $\sqrt[3]{-25}$ | From the following infinite list of numbers, how many are integers? $\sqrt[2]{4096}, \sqrt[3]{4096}, \sqrt[4]{4096}, \sqrt[5]{4096}, \sqrt[6]{4096}, \ldots$ | Rewrite $\sqrt[3]{2^6 \cdot 3^3 \cdot 11^3}$ as an integer. | Evaluate $\lceil\sqrt{5}\rceil + \lceil\sqrt{6}\rceil + \lceil\sqrt{7}\rceil + \cdots + \lceil\sqrt{29}\rceil$ Note: For a real number $x$, $\lceil x \rceil$ denotes the smallest integer that is greater than or equal to $x$. |

*Table 10.* This table shows the in-context examples obtained from the skill-exemplar repository based on skill-matching. We can see that the proposed approach provides relevant in-context examples that illustrate the concepts required by the question.

## 9.2. Grade Level Math Word Problems

We present examples from the GSM8K skill exemplar repository in Table 9.

**Skill Wise Accuracy** We study for which skills the proposed approach is most beneficial in by comparing the per-skill accuracy of the proposed Skill-Based approach against the Random baseline. This comparison is presented in Figure 3. We can see that the proposed approach outperforms the Random Baseline in 11/18 skills.

## 9.3. MATH Dataset: Competition Level Math Problems

We present example from the MATH skill exemplar repository in Tables 14 and 15.

**Number of skill obtained** After the skill labelling phase, we end up with 823 skills for prealgebra, 877 for algebra, 805 for intermediate algebra, 620 for geometry, 492 for number theory, 525 for pre calculus, and 406 for probability. After clustering, we end up with 14 skills for prealgebra, 21 for algebra, 23 for intermediate algebra, 14 for geometry, 15 for number theory, 19 for precalculus, and 11 for probability. Tables 14 and 15 show examples from the skill exemplar repository for the math dataset.

**Examples of relevant in-context examples** In Table 10, we present examples of relevant in-context examples provided by the skill exemplar repository.

## 9.4. Comparing Skill Annotation Models

In this section, we compare GPT-4, GPT-3.5, and Mixtral-8x7B as skill annotators for labelling questions with skills and clustering skills. For skill labeling and clustering we feed the prompts listed in Figure 2 to all the models. We also tested Llama-2 70B for skill annotation but we found that it was not able to provide a sensible skill name for any example. It struggles to understand the instruction given in the prompt in Figure 2 (left). Therefore, we discarded it as the skill annotation model.

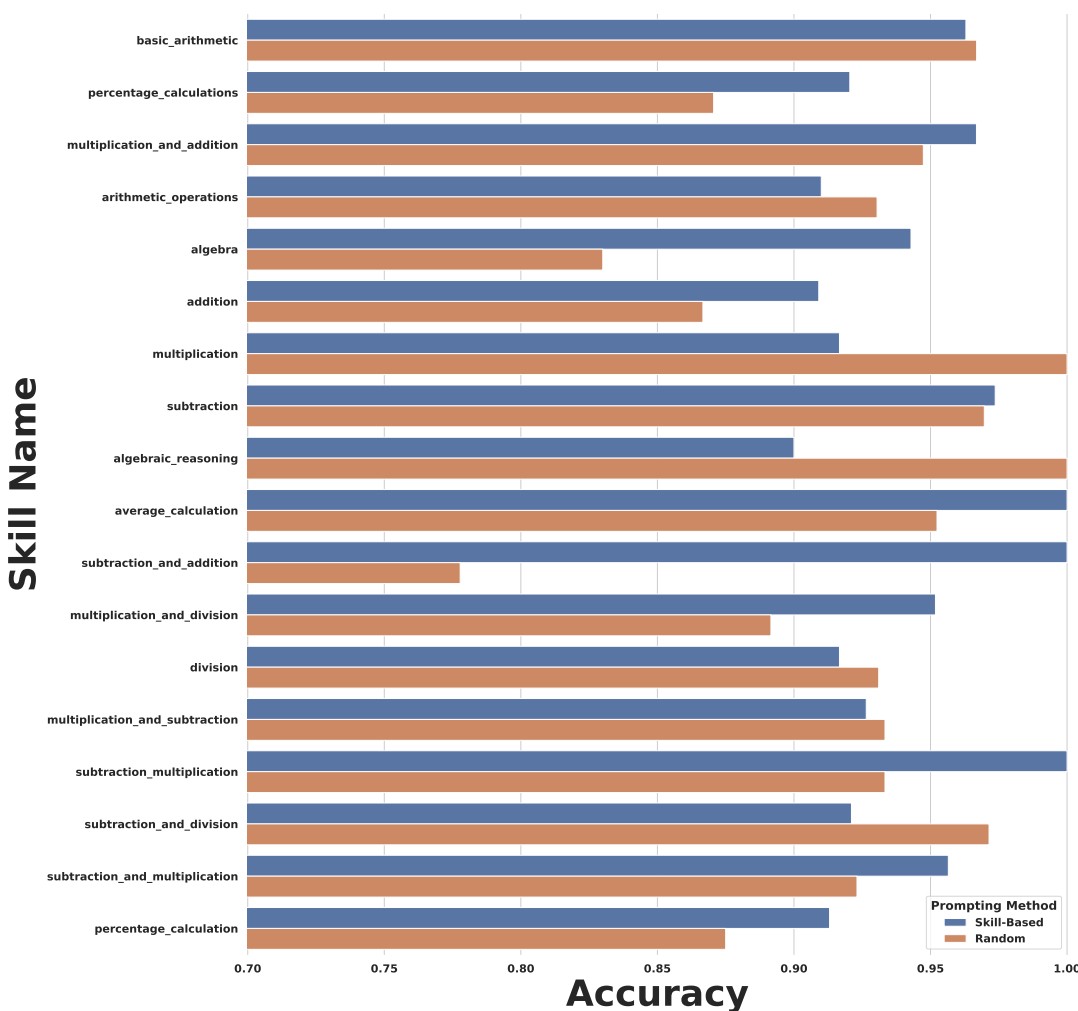

*Figure 3.* **Skill Wise Plot** In this Figure we compare the the per-skill accuracies for the Skill-Based approach and the Random approach on the GSM8K dataset. We can see that proposed Skill-Based approach results in better accuracies for 11/18 skills.

Next, we found that Mixtral-8x7B, GPT-3.5, and GPT-4 are able to label question with skills as expected but GPT-4 was more descriptive and in some cases more accurate as well as shown in Table 11.

| Question | Mixtral-8x7B skill | GPT-3.5 skill | GPT-4 skill |
|---|---|---|---|
| There are positive integers that have these properties: I. The sum of the squares of their digits is and II. Each digit is larger than the one on its left. What is the product of the digits of the largest integer with both properties? | order_of_operations | number_theory | combinatorics_and_number_theory |
| A Senate committee has 5 Democrats and 5 Republicans. In how many ways can they sit around a circular table if each member sits next to two members of the other party? | combinatorics | counting_and_probability | circular_permutation_combinatorics |
| How many different positive integers can be represented as a difference of two distinct members of the set $\{1, 2, 3, 4, \ldots, 16\}$? | counting | counting_and_probability | counting_and_subtraction |

*Table 11.* Skill Labels Assigned by Mixtral-8x7B, GPT-3.5, and GPT-4

Next, we performed skill clustering with all the above 3 models and found that while GPT-4 and GPT-3.5 succeed at clustering, Mixtral fails to perform sensible clustering. It puts all skills in one cluser.

Therefore, we are left with GPT-4 and GPT-3.5 for skill-labeling and skill-clustering. We create two different skill exemplar repositories for GPT-4 and GPT-3.5 respectively. We compare these skill-exemplar repositories by using them to provide relevant in-context examples to solve questions from the MATH dataset. The results for this comparison are presented in Table 12. The superior performance with GPT-4 skills indicates that GPT-4 succeeds at providing higher quality skill annotations as compared to GPT-3.5.

| Topic | Pre-Algebra | Geometry | Inter-Algebra | Algebra | Probability | Pre-Calculus | Num. Theory | Overall |
|---|---|---|---|---|---|---|---|---|
| CoT + Skill-Based (GPT 3.5 skills) | 74.85 | 40.70 | 25.51 | 69.41 | 55.06 | 33.69 | 46.29 | 51.9 |
| CoT + Skill-Based (GPT-4 Skills) | 74.28 | 41.75 | 27.02 | 73.12 | 58.01 | 33.70 | 51.10 | 53.8 |

*Table 12.* In this table, we compare the skill exemplar repositories created using GPT-3.5 and GPT-4 on the MATH dataset. The higher performance with GPT-4 skills illustrates its superiority as the skill annotation model.

## 9.5. Program Assisted Language Models

In this section, we first present the prompt format used in PAL (24):

```
<question 1> <code solution 1>
<question 2> <code solution 2>
<question 3> <code solution 3>
<question 4> <code solution 4>
            <Q>...
```

Next, we show how we modify this prompt format to incorporate skills exemplars from the Skill Exemplar Repository:

```
<question 1> <text solution 1>
<question 2> <text solution 2>
<question 3> <text solution 3>
<question 4> <code solution 4>
            <Q>...
```

Here, `<question 1> <text solution 1> <question 2> <text solution 2> <question 3> <text solution 3>` are exemplars from skill exemplar repository exhibiting the same skill as `<Q>`.

## 9.6. Examples where the Model makes Mistakes despite Receiving Skill Exemplars

In Table 13 we present examples where the model makes mistakes despite receiving in-context examples with the same skill as the question it needs to answer. The discussion regarding each example is presented in the main paper.

|  | Example 1 | Example 2 |
|---|---|---|
| **Question** | *Evaluate* $\log_{\sqrt[3]{5}} 125$. | *How many distinct diagonals of a convex hexagon can be drawn?* |
| **Topic-Based Solution** | The expression $\log_{\sqrt[3]{5}} 125$ asks the question "to what power must we raise $\sqrt[3]{5}$ to obtain 125?". Since $5^{\frac{1}{3}} = 5$, then $5^{3x} = 125$. Recall that $5^3 = 125$. Therefore 125 is actually $5^3$. Hence the $\log_{\sqrt[3]{5}} 125 = 3$. | A hexagon has 6 vertices. A diagonal can be drawn from a vertex to any other vertex except for the two adjacent vertices. So, from each vertex, we can draw 3 diagonals. Therefore, for 6 vertices, we can draw $6 \times 3 = 18$ diagonals. We divide this by 2 to get the number of distinct diagonals. Therefore, a convex hexagon has 9 distinct diagonals. |
| **Skill-Based Solution** | We know that $\log_a b = c$ means $a^c = b$. So we need to find a number which, when cubed, gives us 125. We know that $5^3 = 125$, so our answer is 3. **Skill Used:** logarithmic_and_exponential_skills | A convex hexagon has 6 vertices. From each vertex, we can potentially draw a diagonal to 4 other vertices. So, each vertex gives us 4 diagonals, for a preliminary count of $6 \times 4 = 24$ diagonals. We must divide by 2 to get the number of unique diagonals. The answer is $(6 \times 4)/2 = 12$ diagonals. **Skill Used:** counting_and_number_theory |

*Table 13.* **Incorrectly answered examples despite providing skill-based exemplars** This table highlights situations where the model encounters reasoning errors despite being supplied with relevant skill-based exemplars to address the problem. Correct reasoning steps are denoted in blue, while errors are highlighted in red.

## 9.7. Prompting GPT 4 to classify errors

The prompt which is given to GPT 4 to categorize examples into SKILL ERROR, SECONDARY SKILL ERROR, and CALCULATION ERROR is shown in Figure 5.

(a) **Skill Success Rate**  (b) **Secondary Concept Success Rate**  (c) **Calculation Success Rate**

*Figure 4.* **Ablation Metrics** This Figure compares the SKILL SUCCESS RATE, SECONDARY SKILL SUCCESS RATE, and CALCULATION SUCCESS RATE of the Topic-Based and Skill-Based approaches. We expect the proposed skill-based approach to be mainly useful in picking the correct skills. We find that this is indeed the case as it achieves a higher skill success rate than the Topic-Based approach. Moreover, we find that proposed approach also results in lower calculation and secondary skill errors.

## 9.8. Performance on Ablation Metrics

We present the performance on all the 3 newly introduced metrics in Figure 4. We expect the proposed Skill-Based approach to be the most beneficial in reducing SKILL ERRORS. This is because the the proposed approach should provide those in-context examples to the model which have the same main skill as the problem in question. The performance on the SKILL SUCCESS RATE metric is in-line with this hypothesis. We observe that the proposed approach results in a higher SKILL SUCCESS RATE which means that the model is using the correct skill more frequently in the proposed approach as compared to the Topic-Based baseline. Furthermore, we find that the proposed approach is also quite effective in reducing secondary skill errors and calculation errors. Thus, showing the overall superiority of the proposed approach.

You are a math wizard who knows exactly what mathematical concept to use to solve any math question.
I am going to give you a math question and a solution and the groundtruth answer for that question. You need to answer some questions that I ask you about it.
Here are examples of questions and the corresponding answers:
Question: We call a number a descending number if each digit is strictly smaller than the digit that comes before it. For example, 863 is a descending number. How many 3-digit descending numbers are there?
Solution: Since 0 cannot be the leading digit of the number, there are 9 options for the first digit (1-9). Once the first digit is chosen, there are 10 options for the second digit (0 plus the digits less than the first digit). Both the second and first digits are determined, this leaves 9 options for the third digit (0 and 1-9, excluding the second digit). Hence, there are $9 * 10 * 9 = 810$ possible 3-digit numbers. However, this count includes numbers such as 100, 200, etc – where the third digit is not strictly less than the second digit. There are exactly 9 such numbers. So our final answer is $810 - 9 = \boxed{801}$ descending numbers.
Groundtruth: 120
Q1: Are there any calculation mistakes in the solution?
Q2: Are there any mistakes in the main concept required to solve the question?
Q3: Are there any mistakes in any of the secondary concepts required to solve the question?
A1: no —— There are two calculations in the question: 9*10*9=810 and 810 - 9 = 801. Both are correct.
A2: yes —— The main concept is understanding of descending numbers, but this understanding is not correct.
A3: no —— The secondary concept of permutations is correctly used.
Question: How many primes are in the row of Pascal's Triangle that starts with a 1 followed by a 6?
Solution: The row of Pascal's Triangle that starts with a 1 followed by a 6 is the sixth row, which is composed of the numbers 1, 6, 15, 20, 15, 6, 1. The only prime number in this row is $\boxed{\{2\}}$, which is the number 2 itself.
Groundtruth: 0
Q1: Are there any calculation mistakes in the solution?
Q2: Are there any mistakes in the main concept required to solve the question?
Q3: Are there any mistakes in any of the secondary concepts required to solve the question?
A1: no —— There are no calculations in the solution.
A2: yes —— The main concept is understanding of primes, but this understanding is not correct.
A3: no —— The secondary concept of Pascal's Triangle is correctly used.
Question: Two different numbers are randomly selected from the set $\{1, 2, 3, 4\}$ and they are multiplied. What is the probability that the product is even? Express your answer as a common fraction.
Solution: There are a total of $\binom{4}{2} = 6$ pairs of numbers that can be chosen. The pairs that yield an even product are (1,2), (2,3), (2,4), (1,4), and (3,4) for a total of 5 such pairs. Therefore, the probability is $\frac{5}{6}$. However, each pair can be ordered in two ways (for example, (1,2) and (2,1) are different orderings of the same pair). So we should actually consider that there are 2*6=12 total outcomes and 2*5=10 successful outcomes. Therefore the probability is $\{10\}\{12\} = \boxed{\frac{5}{6}}$.
Groundtruth: 5/6
Q1: Are there any calculation mistakes in the solution?
Q2: Are there any mistakes in the main concept required to solve the question?
Q3: Are there any mistakes in any of the secondary concepts required to solve the question?
A1: no —— There are two calculations in the question: 5/6 and 10/12. Both are correct.
A2: no —— The main concept is understanding of probability, and this understanding is correct.
A3: no —— The secondary concept of combinations is correctly used.
Question: The asymptotes of a hyperbola are $y = 2x - 3$ and $y = 17 - 2x$. Also, the hyperbola passes through the point $(4, 7)$. Find the distance between the foci of the hyperbola.
Solution: The two lines $y = 2x - 3$ and $y = 17 - 2x$ intersect at the point $\left(\frac{20}{4}, \frac{14}{4}\right) = (5, 3.5)$, which is the center of the hyperbola. The slopes of the asymptotes are equal to $\pm\frac{b}{a}$, where $a$ and $b$ are the semi-major and semi-minor axes of the hyperbola, respectively. Therefore, $\frac{b}{a} = 2$, so $b = 2a$. The hyperbola passes through the point $(4, 7)$, so the distance from this point to the center of the hyperbola is $\sqrt{(4 - 5)^2 + (7 - 3.5)^2} = a\sqrt{1 + 2^2} = a\sqrt{5}$. Thus, $a = 1.5$. Then $b = 2a = 3$, so the distance between the foci is $2\sqrt{a^2 + b^2} = 2\sqrt{(1.5)^2 + 3^2} = \boxed{6}$.
Groundtruth: 2
sqrt{5}
Q1: Are there any calculation mistakes in the solution?
Q2: Are there any mistakes in the main concept required to solve the question?
Q3: Are there any mistakes in any of the secondary concepts required to solve the question? A1: no —— There are two calculations in the question: $2 * sqrt(1.5^2 + 3^2)$ and $sqrt(1 + 2^2)$. Both are correct.
A2: no —— The solution uses the correct concept of finding the intersection of the asymptotes to determine the center of the hyperbola, and correctly identifies the slopes of the asymptotes as $\pm\{\frac{b}{a}\}$.
A3: yes —— The solution incorrectly equates the distance from the center of the hyperbola to the point it passes through with $a\sqrt{1 + 2^2} = a\sqrt{5}$. This is incorrect because for a hyperbola, the distance from the center to a point on the hyperbola is not related to the semi-major and semi-minor axes in this way.
Question: <question>
Solution:<solution>
Groundtruth: <ground_truth>
Q1: Are there any calculation mistakes in the solution?
Q2: Are there any mistakes in the main concept required to solve the question?
Q3: Are there any mistakes in any of the secondary concepts required to solve the question?
You should answer Q1, Q2, and Q3 based on the given examples.

*Figure 5.* **Ablation Prompt** This figure shows the prompt which is given to GPT 4 to categorize each example from the MATH dataset into SKILL ERROR, SECONDARY SKILL ERROR, or CALCULATION ERROR

| Topic | Skill | Question | Answer |
|---|---|---|---|
| Pre-Algebra | prime_number_theory | What is the sum of the distinct prime factors of 315? | The distinct prime factors of 315 are 3, 5, and 7, and their sum is 15. |
| | exponentiation_rules | Find $(4^4 \div 4^3) \cdot 2^8$. | Performing the arithmetic in the parentheses first, we obtain $4^4 \div 4^3 = 4$, so we have $(4^4 \div 4^3) \cdot 2^8 = 4 \cdot 2^8$. Since $4 = 2^2$, we have $4 \cdot 2^8 = 2^2 \cdot 2^8 = 2^{10}$ $\boxed{1024}$. |
| | ratio_and_proportion | The ratio of irises to roses in Nikki's garden is 2:5. She currently has 25 roses. She is planning to add 20 more roses and enough irises to keep the same ratio. How many irises will she have in total after this addition? | After adding 20 roses, Nikki will have $25 + 20 = 45$ roses. If she will have 2 irises for every 5 roses, then she will have $2(9) = \boxed{18}$ irises. |
| Algebra | factoring_skills | Factor $9y^2 - 30y + 25$. | The quadratic is the square of $3y$, the constant term is the square of $-5$, and the linear term equals $2(3y)(-5)$, so we have $9y^2 - 30y + 25 = \boxed{(3y - 5)^2}$. |
| | complex_number_skills | Simplify $(3 - 2i)^2$. (Your answer should be of the form $a + bi$.) | $(3 - 2i)^2 = (3 - 2i)(3 - 2i) = 3(3) + 3(-2i) - 2i(3) - 2i(-2i) = 9 - 6i - 6i - 4 = \boxed{5 - 12i}$. |
| | quadratic_equation_skills | What is the sum of the values of $x$ that satisfy the equation $x^2 - 5x + 5 = 9$? | Subtracting 9 from both sides of the equation, we have $x^2 - 5x - 4 = 0$. The sum of the roots of this quadratic is negative its linear coefficient, which is $\boxed{5}$. (The above is true because if a quadratic has roots $r$ and $s$, we have $(x-r)(x-s) = x^2 - (r+s) + rs = 0$.) |
| Intermediate Algebra | polynomial_skills | Find the product of the nonreal roots of $x^4 - 4x^3 + 6x^2 - 4x = 2005$. | We recognize part of the expansion of $(x - 1)^4$ on the left-hand side. Adding 1 to both sides, we have $x^4 - 4x^3 + 6x^2 - 4x + 1 = 2006$, which means $(x - 1)^4 = 2006$. Therefore, $x - 1 = \sqrt[4]{2006}, i\sqrt[4]{2006}, -\sqrt[4]{2006}, -i\sqrt[4]{2006}$. Since we want the nonreal roots, we only consider the roots $$x = 1 \pm i\sqrt[4]{2006}.$$ The product of these roots is $P = (1 + i\sqrt[4]{2006})(1 - i\sqrt[4]{2006}) = \boxed{1 + \sqrt{2006}}$. |
| | simplifying_and_basic_operations | Let $P = \left(1 - \frac{1}{2}\right)\left(1 - \frac{1}{3}\right)\left(1 - \frac{1}{4}\right) \cdots \left(1 - \frac{1}{n}\right)$. What is the value of $P$ if $n = 2007$? Express your answer as a common fraction. | Simplifying each term in $P$, $$P = \left(\frac{1}{2}\right)\left(\frac{2}{3}\right)\left(\frac{3}{4}\right) \cdots \left(\frac{n-1}{n}\right).$$ The denominator of each fraction cancels with the numerator of the next fraction, so $P = \frac{1}{n}$. When $n = 2007$, $P = \boxed{\dfrac{1}{2007}}$. |
| | graph_understanding_and_interpretation | Find the distance between the vertices of the hyperbola $\frac{x^2}{99} - \frac{y^2}{36} = 1$. | We read that $a^2 = 99$, so $a = \sqrt{99} = 3\sqrt{11}$. Therefore, the distance between the vertices is $2a = \boxed{6\sqrt{11}}$. |
| Geometry | pythagorean_skills | In right triangle $ABC$, $AB = 10$, $AC = 6$ and $BC = 8$ units. What is the distance from $C$ to the midpoint of segment $AB$? | The length of the median to the hypotenuse of a right triangle is half the length of the hypotenuse. Therefore, the desired distance is $10/2 = \boxed{5}$. |
| | 3d_geometry_and_volume_calculation_skills | The area of one lateral face of a right pyramid with an equilateral triangular base is 75 square meters. If the slant height is 30 meters, what is the length of the side of its base, in meters? | Let $s$ represent the sidelength of the equilateral triangular base. Each face of the pyramid has an area of $\frac{1}{2}bh = 75$, where $b$ is the sidelength of the base and $h$ is the slant height of 30 meters. We have $$75 = \frac{1}{2}s(30) = 15s.$$ So, $s = 5$ and the sidelength of the base is $\boxed{5}$ meters. |
| | triangle_geometry_skills | In triangle $ABC$, $AB = 16$, $AC = 24$, $BC = 19$, and $AD$ is an angle bisector. Find the ratio of the area of triangle $ABD$ to the area of triangle $ACD$. (Express your answer as a fraction in lowest terms.) | The ratio of the area of triangle $ABD$ to the area of triangle $ACD$ is $BD/CD$. By the angle bisector theorem, $BD/CD = AB/AC = 16/24 = \boxed{\dfrac{2}{3}}$. |

*Table 14.* **Math skill exemplar repository** This table presents few examples from the skill exemplar repository for 5 topics from the MATH dataset (16).

| Topic | Skill | Question | Answer |
|---|---|---|---|
| Precalculus | calculus | If $\tan\alpha = 8$ and $\tan\beta = 7$, then find $\tan(\alpha - \beta)$. | From the angle subtraction formula, $\tan(\alpha - \beta) = \frac{\tan\alpha - \tan\beta}{1+\tan\alpha\tan\beta} = \frac{8-7}{1+8\cdot7} = \boxed{\frac{1}{57}}$. |
| | vector_operations | Find $y$ so that the vectors $\begin{pmatrix}1\\-3\\-4\end{pmatrix}$ and $\begin{pmatrix}-2\\y\\-1\end{pmatrix}$ are orthogonal. | For the vectors $\begin{pmatrix}1\\-3\\-4\end{pmatrix}$ and $\begin{pmatrix}-2\\y\\-1\end{pmatrix}$ to be orthogonal, their dot product should be 0: $(1)(-2) + (-3)(y) + (-4)(-1) = 0.$ Solving, we find $y = \boxed{\frac{2}{3}}$. |
| | trignometric_calculations | Convert $e^{11\pi i/2}$ to rectangular form. | We have that $e^{11\pi i/2} = \cos\frac{11\pi}{2} + i\sin\frac{11\pi}{2} = \boxed{-i}$. |
| Number Theory | factorization | Find the product of the divisors of 50. | For every divisor $d$ of 50, then $50/d$ is also a divisor of 50. Their product is $d \cdot (50/d) = 50$. It follows that every divisor of 50 can be paired with another divisor of 50 such that their product is $50 = 2\cdot5^2$. There are $(1+1)(2+1) = 6$ divisors of 50: 1, 2, 5, 10, 25, 50. Thus, the answer is $50^{6/2} = 50^3 = \boxed{125{,}000}$. |
| | division_and_remainders | A whole number is said to be "9-heavy" if the remainder when the number is divided by 9 is greater than 5. What is the least three-digit 9-heavy whole number? | We begin by computing the residue of the smallest three digit number modulo 9. We have $$100 \equiv 1 \pmod 9.$$ Therefore 100 is not 9-heavy. Counting up from 100 we notice that the first 9-heavy three-digit number is $\boxed{105}$, since it has a remainder of 6 when divided by 9. |
| | exponentiation | Call an integer $n$ oddly powerful if there exist positive integers $a$ and $b$, where $b > 1$, $b$ is odd, and $a^b = n$. How many oddly powerful integers are less than 2010? | Let us first determine the number of cubes that are less than 2010. We have $10^3 = 1000$, $11^3 = 1331$, and $12^3 = 1728$, but $13^3 = 2197$. So there are 12 cubes less than 2010. As for fifth powers, $4^5 = 1024$, but $5^5 = 3125$. There are 4 fifth powers less than 2010, but only 3 of these have not already been included, since we've already counted 1. Analyzing seventh powers, $3^7 = 2187$, so the only new seventh power less than 2010 is $2^7$. There are no new ninth powers since they are all cubes, and $2^{11} = 2048$ is greater than 2010. Therefore, there are $12+3+1 = \boxed{16}$ oddly powerful integers less than 2010. |
| Probability | combinatorics_knowledge | Alex has 10 different kinds of lunch meat and 9 different kinds of cheese. If he wants to make a sandwich with one kind of meat and two kinds of cheese, how many different sandwiches could he make? (It does not matter in which order he chooses the two types of cheese.) | There are $\binom{10}{1} = 10$ ways for Alex to choose which kind of lunch meat to put on his sandwich, and there are $\binom{9}{2} = 36$ ways for Alex to choose which kinds of cheese to put on his sandwich. The total number of different sandwiches Alex can make is $10 \cdot 36 = \boxed{360}$. |
| | permutations_and_combinations | A bag contains 10 red marbles and 6 blue marbles. Three marbles are selected at random and without replacement. What is the probability that one marble is red and two are blue? Express your answer as a common fraction. | There are three ways to draw two blue marbles and a red one: RBB, BRB, and BBR. Since there are no overlapping outcomes, these are distinct cases and their sum is the total probability that two of the three drawn will be blue. The desired probability therefore is $$\frac{10}{16}\cdot\frac{6}{15}\cdot\frac{5}{14} + \frac{6}{16}\cdot\frac{10}{15}\cdot\frac{5}{14} + \frac{6}{16}\cdot\frac{5}{15}\cdot\frac{10}{14} = \boxed{\frac{15}{56}}.$$ |
| | counting_princpals | How many three digit numbers are there? | The three-digit numbers start with 100 and end with 999. There are $999-100+1 = \boxed{900}$ three-digit numbers. |

*Table 15.* **Math skill exemplar repository (Continued)** This table presents few examples from the skill exemplar repository for 5 topics from the MATH dataset (16).

