# OpenReview forum: "Metacognitive Capabilities of LLMs: An Exploration in Mathematical Problem Solving"
_ICML.cc/2024/Workshop/AI4MATH — ICML 2024 Workshop AI4MATH Poster_

### Official Review · Reviewer_duKp · 2024-06-11

**Rating:** 8
**Confidence:** 4

**Summary:**

Metacognitive knowledge pertains to the innate understanding humans possess regarding their own thinking and reasoning processes. Presently, state-of-the-art Large Language Models (LLMs) demonstrate evident reasoning abilities. The paper presents compelling evidence suggesting that these models also possess metacognitive knowledge, enabling them to identify and employ appropriate skills and procedures when presented with a given task.

**Questions:**

For Table 2., what is the result of Complex CoT+ Skill-Based?

**Reasons To Accept:**

**Originality:** The Metacognitive knowledge analysis is original and new.

**Quality:** The authors conduct extensive experiments to prove the claims.

**Clarity:** The paper is well-written. And it is easy to understand the paper.

**Significance:** It is important to analyze the metacognitive knowledge and build skill exemplars.

**Reasons To Reject:**

Is the method suitable for other reasoning, such as commonsense reasoning? Or, is the method limited to math reasoning?

---

### Official Review · Reviewer_nVJu · 2024-06-12

**Rating:** 7
**Confidence:** 4

**Summary:**

The authors considered metacognitive capabilities of LLMs which is relatively new but worthy further studies; the experiments method proposed can be regarded a good benchmark for future research. Thus, it is acceptable as a poster at least.

**Questions:**

1. Have the authors tried to check if there are any other similar research on this topic? If there are any, did you do a comparisons?

2. How to make sure that the results of the experiments are solid evidence of LLMs' metacognitive capabilities rather than some similarity patterns discovered by LLM which are hard for human to understand?

3. Have you tested the generalization abilities of the metacognitive capabilities of LLMs?

**Reasons To Accept:**

The topic considered in this paper is relatively new and there are not many researches on this direction yet; thus, the experiment proposed in it is good to be introduced to a wider audience.

**Reasons To Reject:**

At least it is good enough for a poster; the only concerns are there are not many good benchmarks to compare the proposed experiment in this paper, and there seems no standard way to check/measure if what the authors found is exactly LLMs' metacognitive capabilities.

---

### Official Review · Reviewer_tgUo · 2024-06-12

**Rating:** 5
**Confidence:** 4

**Summary:**

This paper investigates Metacognitive Knowledge in Large Language Models (LLMs), specifically focusing on the ability to name skills and apply them to solve novel questions within the context of mathematical reasoning.

The authors propose a framework that requires GPT-4 to initially discover skills from a training set, cluster these skills into coarser grained categories, and finally build a skill exemplar repository. Each data point in the training set is associated with specific skills.

At test time, the framework prompts GPT-4 to identify relevant skills from the repository for a new question, and selects examples based on these skills for few-shot prompting. This method integrates text-based and program-aided prompting, and explores the transfer of skill exemplars to other datasets and models.

The experimental results demonstrate improvements attributed to the framework.

**Questions:**

Please refer to Reasons to Reject.

**Reasons To Accept:**

- Originality: The link between skill identification, application, and Metacognitive Knowledge offers a novel perspective.
- Clarity: The framework is conceptually simple and clearly described.

**Reasons To Reject:**

- Originality: The skills discussed (e.g., understanding triangle properties and circle radius calculations) closely resemble the knowledge points explored in KPMath[1] and MathScale[2]. These works have previously proposed the creation of large-scale synthetic data, which resulted in significant improvements in mathematical reasoning.
- Clarity: The organization of the paper could be improved. The related works section is inadequately articulated, and the experiments section conflates the setup with the main results, complicating comprehension.
- Effectiveness: The improvement noted in this paper is unremarkable. For instance, PAL + Skill-Based (7 Skill-Based + 1 Code-Based) achieves a 62.0% accuracy on the MATH dataset using GPT-4, whereas previous code-based research[3] achieved a substantial increase in zero-shot accuracy on the same dataset (from 53.9% to 84.3%) using GPT-4.

[1] Huang, Yiming, et al. "Key-Point-Driven Data Synthesis with its Enhancement on Mathematical Reasoning." arXiv preprint arXiv:2403.02333 (2024).

[2] Tang, Zhengyang, et al. "Mathscale: Scaling instruction tuning for mathematical reasoning." arXiv preprint arXiv:2403.02884 (2024).

[3] Zhou, Aojun, et al. "Solving challenging math word problems using GPT-4 code interpreter with code-based self-verification." arXiv preprint arXiv:2308.07921 (2023).

---

### Meta-Review · Area_Chair_KWNR · 2024-06-13

**Recommendation:** Accept (Poster)
**Confidence:** 4

**Metareview:**

Acceptable paper. The paper proposes a novel method for skill identification and application to enhance GPT-4's mathematical reasoning capabilities. The experiments demonstrate a noticeable improvement. Metacognitive knowledge offers a new perspective on the skill-based approach. While the method is innovative, it is not significantly groundbreaking, given the numerous existing works that focus on learning and applying skills in the domain of AI for mathematics.

---

### Decision · Program_Chairs · 2024-06-13

Accept (Poster)